# Recent Advancement in Commercial and Other Sustainable Techniques for Energy and Material Recovery from Sewage Sludge

Mohd Imran Siddiqui [image: ORCID], Hasan Rameez [image: ORCID], Izharul Haq Farooqi and Farrukh Basheer *

Department of Civil Engineering, Zakir Husain College of Engineering and Technology,
Aligarh Muslim University, Aligarh 202002, India
* Correspondence: farrukhbasheer.cv@amu.ac.in

**Abstract:** Rapid population growth and urbanization have resulted in a multi-fold increase in water consumption over the last few decades, resulting in the generation of large amounts of sewage and sewage sludge that impose severe environmental burdens if not handled properly. Sludge management itself accounts for up to 50% of the total operating costs of wastewater treatment plants (WWTPs). Conventional sludge management practices such as incineration, landfilling, and ocean disposal have been deemed difficult in light of today's stringent environmental legislation and compliance standards. As a result, progress has been made toward developing more sustainable approaches for sludge management. This study reviews recent advancements in sewage sludge management techniques that not only ensure the safe disposal of sewage sludge but also focus on utilizing the potential of sewage sludge as feedstock for energy and resource recovery. Energy could be recovered by subjecting the pre-treated sludge to controlled anaerobic digestion (AD) to produce biogas or by utilizing the lipid content of the sewage sludge through esterification or direct sludge pyrolysis to produce biodiesel/bio-oil. Heavy metals such as Ag, Au, Cu, Fe, Ga, Cr, and others, as well as nutrients such as N, P, K, Mg, S, and others, could also be recovered. If energy and resource recovery from sewage sludge is practiced on a global scale, it could significantly contribute to global greenhouse gas (GHG) emission reduction. This review discusses the commercially developed and still-under-research technologies for energy and other resource recovery of sewage sludge. Additionally, techniques, along with their limitations and potential measures to improve their yields, are also discussed.

**Keywords:** anaerobic digestion; biogas; biodiesel; heavy metal recovery; nutrient recovery; sewage sludge; sludge pre-treatment; sustainable management

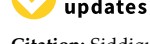



## 1. Introduction

The management of sewage sludge is a global concern. The world's population is expected to grow up to 9.8 billion by 2050 and 11.2 billion by 2100 [1]. Population growth and rapid urbanization results in an increase in sewage generation. This sewage typically contains greywater and blackwater from domestic households and in some cases could even include industrial effluents as a result of discharge into public sewers. This necessitates the need for an increased number of wastewater treatment plants (WWTPs). As a result, the amount of sewage sludge generated by WWTPs has increased, negatively impacting the ecosystem [2]. As per a report by the European Investment Bank (2022) [3], total municipal wastewater generation around the globe is 380 billion m³/year and is expected to increase by 24% and 51% in 2030 and 2050, respectively. The European Union produces approximately 50 million tons of sewage sludge each year, while the United States produces 40 million tons, and China produces 60 million tons of sewage sludge (80% water by weight) with an annual increase rate of 10% [4]. One study revealed that the average per capita generation of sewage sludge is 70 g/day (dry solid basis) [5] and

the energy potential of this sludge is estimated to be in the range of 11.5 to 16 kJ/g (dry solid basis) [6]. Another study estimated that sewage sludge management accounts for as much as 50% of the total operating cost of the WWTPs [7]. Historically, sewage sludge was handled in a number of ways, such as incineration, ocean disposal, landfilling, and agricultural use [8]. However, over the years these techniques have been found to pose serious environmental threats. The London Convention 97 protocol prohibits discharge of sludge in the oceans as a measure to protect marine ecosystems [9]. Furthermore, the presence of organic contaminants, pathogens, heavy metals, and other trace components limit the use of sludge as a fertilizer by land application of raw sludge [10,11]. Because of the limited availability of land resources and the risk of environmental harm, new laws and regulations prohibit sludge landfilling [12].

Population growth will need a massive quantity of fuel to meet energy demands in the future [13]. A British Petroleum report [14] estimated that global oil consumption will be 1.6 Mb/d (million barrel/day), while global oil generation will increase by only 0.4 Mb/d (million barrel/day). Furthermore, the world's known oil reserves were projected to be 1707 Bb (billion barrel) in 2016, limiting oil production to 50.6 years [14]. This energy deficit must be addressed with new sustainable and cleaner sources of energy that can reduce atmospheric greenhouse gas (GHG) emissions, which are primarily driven by fossil fuels and industrial activity [15]. Today, alternative sustainable fuels are a critical and die-hard need of the time. Sustainable alternative fuels differ from conventional fossil fuels as they are renewable, cost-effective, efficient, and have a lower environmental impact [16].

In the light of the above discussion, researchers are constantly attempting to develop effective techniques for utilizing sludge as a resource. Sewage sludge has recently been recognized as a renewable source of energy and material recovery, giving it the distinction of being a carbon-neutral material [2]. Using sewage sludge as a feedstock is considered advantageous due to its ample and steady availability. The potential for nutrient recycling and energy production of sewage sludge is expected to improve with increases in population, income, and dietary nutritional density [17]. According to [18], if the world's total sewage and sewage sludge were collected and digested anaerobically, it could generate 210–300 TWh of electricity. This amount of energy is enough to meet Ukraine's natural gas needs or the electricity needs of 27 to 38 million people. Furthermore, AD of this quantity of sludge could reduce GHG emissions by 75 to 100 Mt $CO_2$ eq per year, which is comparable to Israel's emissions.

Nowadays, sewage sludge is emerging as an appealing renewable source of energy recovery in biorefineries as well as a potential solution to reduce GHG emissions [19]. Energy could be recovered by subjecting the pre-treated sludge to controlled anaerobic digestion (AD) to produce biogas or by utilizing the lipid content of the sewage sludge through esterification or direct sludge pyrolysis to produce biodiesel/bio-oil. Biofuel (biogas/biodiesel/bio-oil, etc.) is gaining popularity as an alternative to mineral diesel derived from traditional fossil sources. Biofuel could not compete with conventional fossil fuels due to the significantly higher production costs resulting from very expensive feedstock material. Significant research is being conducted globally with the prime focus on different approaches towards economical production of biofuels from the renewable biomass to reduce dependence on conventional fossil fuels. Sewage sludge with high lipid content is a potential feedstock for biofuel synthesis due to its low cost and ample availability [20].

Heavy metals such as Ag, Au, Cu, Fe, Ga, Cr, and others, as well as nutrients such as N, P, K, Mg, S, and others, could also be recovered from the sewage sludge. The presence of heavy metals in sewage sludge is primarily due to the intrusion of industrial wastewater streams into sewage, sewer line deterioration, surface runoff, and other reasons [21,22]. As a number of heavy metals are known to be carcinogenic [23], their presence in sewage and sewage sludge has become a serious threat. Their accumulation in the water and soil impact human health directly or indirectly via the food chain [24]. Although heavy metal recovery potential of sewage sludge is quite significant, carrying out this extraction

process is very challenging economically [25]. It is important to note that the sewage sludge components were valued at 480 USD/dry ton (compared to 7 USD/dry ton for phosphorus), with Cu, Fe, Zn, Pd, Ti, Ir, Cr, Ga, Mn, Au, Cd, Al, and Ag having the greatest potential for economic recovery [26]. Sewage sludge is also valuable in terms of its nutrient content. It is exceptionally high in phosphorus (10–15 wt%), which is important for plant development [27]. After undergoing necessary processing it can be used as an organic fertilizer and soil conditioner in agriculture as it improves soil nutritional profile (NPK and micronutrients) as well as physical properties (lowering bulk density, increasing water retention, and improving cation exchange capabilities) [28]. Given the principles of a circular economy, using it in agriculture is the most environmentally friendly and economically beneficial option [29]. When these nutrients are discharged into water bodies, they cause eutrophication, which degrades water quality by promoting algae growth [30,31]. If all the sewage and sewage sludge are digested anaerobically, 0.23 billion tons of digestate would be produced, containing 2.6 million tons of N, 1.3 million tons of P, 0.12 million tons of K, 0.37 million tons of Mg, and 1.9 million tons of S. This is enough to fertilize 30 million hectares of arable land, replacing approximately 0.4–3% of the world's fertilizer (inorganic) consumption [18].

This paper explores various approaches that have the potential to contribute to sustainable sludge management through the enhancement of energy and resource recovery from sewage sludge. This review provides an in-depth analysis and discussion on a number of cutting-edge methods that are either already being implemented on a commercial scale or hold great promise for such applications in the near future. Additionally, factors affecting the efficiency of these processes and challenges for commercializing them are also discussed.

## 2. Biogas/Methane Production from Sewage Sludge

With the growing demand of energy, biogas recovery from growing sewage streams has become an essential and critical topic. As sludge production in large quantities is inevitable during sewage treatment, it is required to be managed in efficient and sustainable ways. The growth in sewage sludge presents an opportunity to reclaim the biogas which otherwise would have been lost. To achieve this, an efficient sewage treatment system will be required with an advance sludge management infrastructure for biogas recovery. Sludge management, if not conducted efficiently, could alone account for up to 40% of the plant's greenhouse gas emissions [32]. AD has been used for many years for volume reduction and stabilization of sludge, resulting in the production of biogas. As a result, a significant portion of the sludge's GHG emission potential is mitigated in closed systems, and the biogas can be used for beneficial purposes.

According to [18], approximately 39% of the world's population has access to sanitation, indicating that only 39% of potential global sewage is processed in WWTPs or septic tanks and safely disposed of. There are no global statistics on the amount of sewage sludge treated by AD. Assuming that 25% of the sewage sludge collected (from 39% of the population) would be digested using AD, it would represent a net 10% of the world population's potential production. Given that 55% of the world's population resides in metropolitan areas, it is assumed that sewage from these areas can be collected for AD. Rural residents who do not have access to centralized sewers may benefit from bio-toilets with septic tanks or digesters. If even half of the remaining 45% of the population could be linked to proper sanitation facilities, there is potential to attain 77.5% sewage sludge capture. In order to achieve this potential capture of 77.5% by 2050, a targeted capture of 35% should be achieved by 2030. Figure 1 depicts sewage energy production based on a current capture rate of 10% and a prospective generation capacity of 35% by 2030 and 77.5% by 2050. Based on the same assumptions, Figure 2 depicts the potential GHG abatement from energy and nutrient recovery in sewage sludge.

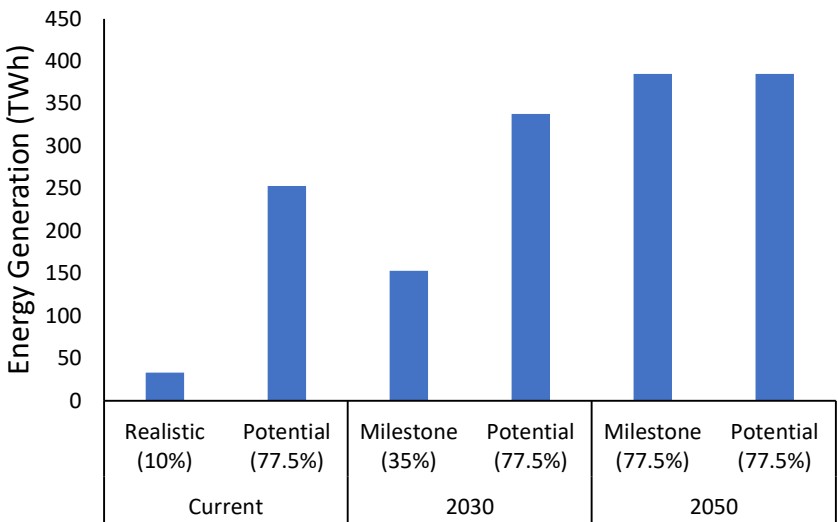

**Figure 1.** Realistic and potential energy generation from sewage sludge [18].

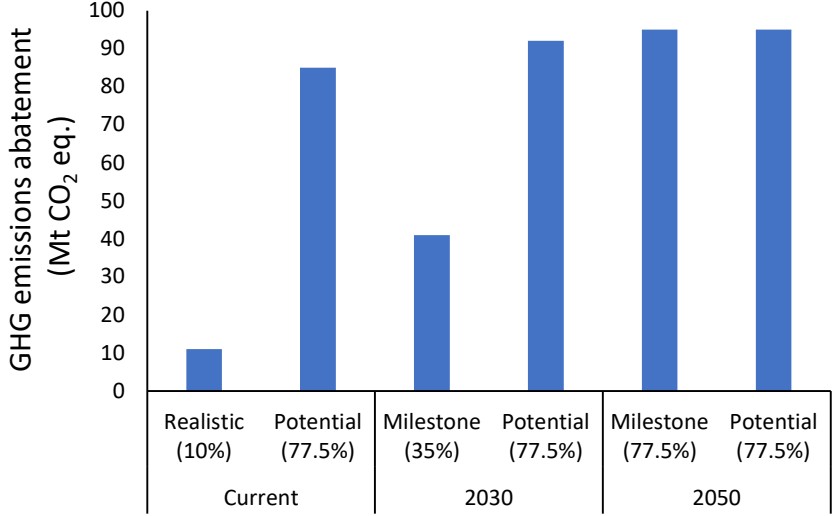

**Figure 2.** Realistic and potential greenhouse gas generation abatement from sewage sludge [18].

This section provides a comprehensive overview of recently developed techniques that are either already in commercial use or that have the potential to be used in the near future. The current status of the global biogas market is also articulated.

*2.1. CAMBI® Technology*

CAMBI® technology is a more recent and advanced treatment method that employs thermal hydrolysis to improve sludge digestion and quality [33]. For over two decades, Cambi's Thermal Hydrolysis Process (THP) has been in use globally to facilitate effective sludge management [34]. The first full-scale pilot-plant was installed in Germany in 2014 and has been continuously in operation since 2015. A study analyzed its operations and reported a 75% reduction in volatile solids (VS) over the sludge treatment line, 43% increased biogas production, and a 69% reduction in wet tons of cake leaving the plant [35].

As shown in Figure 3, CAMBI® is a thermal hydrolysis-based process that primarily consists of three components. A preheated-tank is the first unit that prevents heat exchanger corrosion and facilitates pumping under pressure; a steam-reactor is the second unit where pressurized steam is delivered; and a flash-tank is the third unit where the pressure gets released suddenly. The sludge is dewatered into 14–18% dry solids before being fed into

the preheated-tank. Continuous feeding of pre-dewatered sludge is maintained in the preheated-tank where the sludge temperature is raised over 100 °C. All the heat released from the second and third reactors are guided to the first reactor for economical use of energy and simultaneously maintaining the temperature of the preheated tank. Steam at 12 bar pressure is supplied to the steam reactor for 20–30 min to maintain the temperature and pressure around 150–160 °C and 8–9 bar, respectively. The steam-reactor pressure is reduced to 2 bar before discharging the heated sludge into the flash-tank, and the released steam is recirculated to the preheated-tank. During the discharge of sludge from the steam-tank to the flash-tank, the pressure change is abrupt, which results in the breaking down of the cell wall in the flash-tank. The temperature of the sludge in the flash-tank is around 100 °C, which is brought down to 35 °C before undergoing AD.

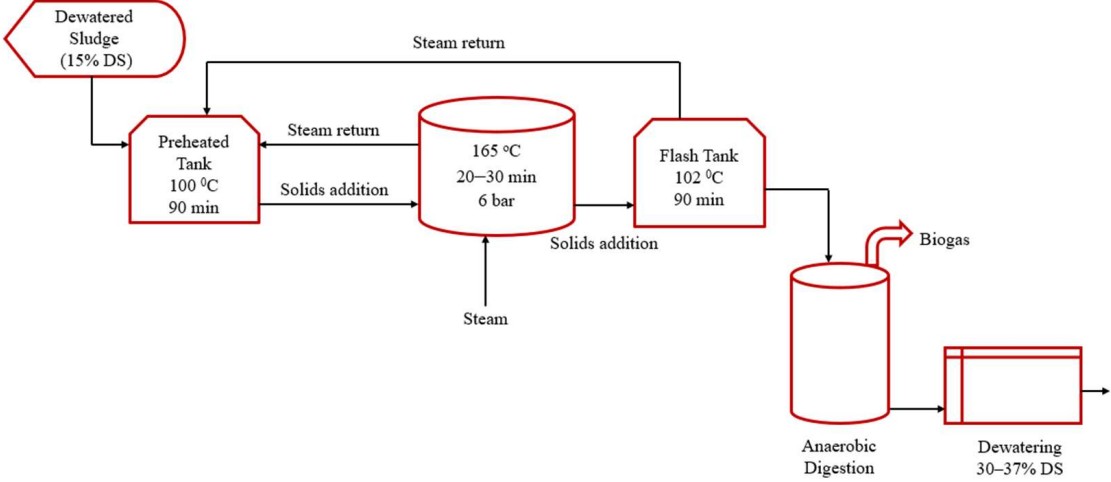

**Figure 3.** Cambi® process (thermal hydrolysis) flow diagram [36].

The CAMBI® process reduces the time required for AD from 15–30 days (conventional) to 10–12 days. Furthermore, the digester volume requirement is cut to half, biogas production increases, and sludge dewaterability is improved (30–40% dry solids content can be achieved) [34]. Furthermore, the technology also minimizes the problems of corrosion, scaling, and residual chemical oxygen demand (COD) in the filtrate that was difficult to decompose. Information relevant to the CAMBI® methodology can be found in the European patent with the number EP0784504 [37].

### 2.2. Bio Thelys^{TM} Technology

Veolia Water Solutions and Technologies spent ten years developing the thermal hydrolysis process known as Bio Thelys^{TM}, followed by anaerobic biological treatment [38]. The primary goals of this technology are to increase the biodegradability of sludge by making the organic content more soluble and to enhance the biogas production in the anaerobic digester. The process primarily comprises a hydrolysis reactor (as shown in Figure 4) into which dewatered sludge with 15–16% dry solids is fed. The sludge is then heated for 30–60 min using direct steam injection, and the temperature within the reactor eventually reaches 150–180 °C. The hydrolyzed sludge is removed from the reactor using the residual pressure, and it is then cooled to the temperature of the anaerobic digester (35 °C). Multiple hydrolysis batch reactors are run in parallel to recover the energy. The flash steam produced during this process is then used to pre-heat another hydrolysis reactor. The hydrolyzed sludge has better dewaterability and lower viscosity, making it easier to pump. The AD of hydrolyzed sludge results in up to 80% more volume reduction as compared to AD of conventional unhydrolyzed sludge, increased biogas production, reduced retention time, smaller digester volume requirement, continuous operation, and process safety [36].

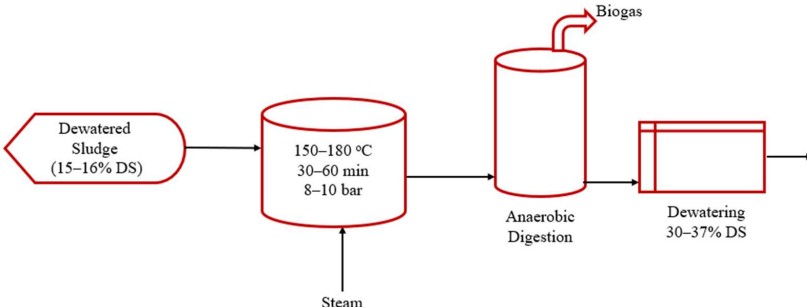

**Figure 4.** Thermal hydrolysis flow diagram for the Bio Thelys^TM process [36].

*2.3. Anaerobic Dynamic Membrane Reactor (AnDMBR) Technology*

The Anaerobic Dynamic Membrane Reactor (AnDMBR) technology for high efficiency sludge digestion and biogas recovery has been successfully implemented in China. A pilot-scale AnDMBR system, which combined anaerobic digestion with dynamic membrane (DM) separation, was used to investigate enhancement in sludge digestion (i.e., employing the produced cake layer on coarse support for solid-liquid separation). The anaerobic dynamic membrane bioreactor, or AnDMBR, is a device that has been extensively researched and advocated for use in the treatment of sewage and sludge in their most common forms [39]. This pilot-scale reactor successfully thickened the sludge and digested it by decoupling the hydraulic retention time (HRT) from the solids retention time (SRT). The total solids concentration in the AnDMBR reached 56.0 g/L after 10 days of HRT and 50 days of SRT, with an influent total solids concentration of 22.8 ± 1.4 g/L. At the same WWTP in Shanghai, the average biogas output of traditional AD was 0.59 L/g VS, and the average vs. reduction rate was 43.9%. While in the AnDMBR-based reactor, the specific biogas production and vs. reduction reached significantly higher values of 0.79 L/g vs. and 55.9%, respectively. The concentration of the digested sludge was more than twice the conventional AD. Additionally, the footprint of the AnDMBR technology is reported to be only 0.3–0.4 times that of conventional AD [40].

*2.4. Current Status of the Biogas Industry*

Even though biogas technologies are widely used across the globe, the sector is still in its growing stage [41]. This section presents the current status of the industry and the use of associated technologies. The biogas sector is divided into three categories: (1) micro-scale digesters that use biogas, (2) scale digesters that generate power, and (3) scale digesters that produce biomethane.

2.4.1. Micro Scale Digesters

Although the usage of micro-scale digesters can be traced back many centuries, it was the seventeenth century that saw the beginning of more recognizable modern applications of biogas reactors. The importance of micro-scale digesters is significant, particularly in rural areas of less developed countries, where they play a critical role in farming, waste management, and energy security. According to [42], there are around 50 million micro-scale digesters operating around the globe, with 42 million located in China and 4.9 million in India. It is estimated that 700,000 biogas plants have been built across the rest of Asia, Africa, and South America. Biogas produced by micro-scale digesters is often used in stoves for cooking and heating, replacing solid fuels such as charcoal and firewood that produce high emissions. About 126 million people worldwide use 50 million stoves powered by biogas to prepare their meals, most of whom live in China (112 million) and India (10 million). Comparatively, digester plants in China produced 13 million m$^3$ of biogas for cooking in 2016, whereas India only produced 2 million m$^3$.

### 2.4.2. Scale (Small, Medium and Large) Digesters Generating Electricity

Biogas-to-electricity generation is a proven technology that has been extensively used worldwide. This is typically accomplished using a combined heat and power (CHP) engine with thermal energy recovery and utilization. A CHP engine is connected to an anaerobic digester in operation. A CHP-based engine must be compact enough for it to be cost-effective. By boosting the use of heat, operators of biogas facilities seek to maximize efficiency and revenue. In this context, trigeneration, which creates power, heat, and cooling as required, is gaining in global popularity.

In addition to the millions of micro digesters, China has a total of 110,448 scale digesters; of them, 6972 are from large-scale facilities (2015). A total of 17,783 scale digesters with the installed capacity of 10.5 GW were found throughout Europe (2017). Germany is the most dominant nation in the European market, with 10,971 plants, followed by Italy with 1655, France with 742, Switzerland with 632, and the United Kingdom with 613. There are 2200 anaerobic digesters in service in the United States, with a total installed capacity of 977 MW [18]. It is anticipated that India has an installed capacity of 300 MW for biogas. Approximately 180 digesters with a total installed capacity of 196 MW are installed in Canada. According to these numbers, around 132,000 digesters of varying sizes are operating worldwide. These numbers include small, medium, and large-scale facilities. The biogas market is growing in many countries throughout the world. According to the latest findings published by the International Renewable Energy Agency [43], the amount of energy globally produced from biogas climbed from 46,108 GWh in 2010 to 87,500 GWh in 2016. That equates to 90% growth in a span of six years.

### 2.4.3. Medium to Large Scale Digesters Upgrading to Biomethane

Converting biogas into biomethane at the commercial level is a relatively new process that has gained widespread acceptance. While some facilities convert biogas into fuel for automobiles, others pump it into local or national networks to be used for heating or cooking. Additionally, greenhouses and other sectors that produce food and beverages are beginning to take advantage of plants' ability to absorb carbon dioxide ($CO_2$). More than 540 upgrading plants can be found throughout Europe, the majority of which are located in Germany (195), the United Kingdom (92), Sweden (70), France (44), and the Netherlands (34), respectively. Outside of Europe, there may be as many as 50 in the United States, 25 in China, 20 in Canada, and a few in Japan, South Korea, Brazil, and India combined. According to the most recent available data, it is reported that 700 facilities throughout the globe convert biogas to biomethane. There are around 344,000 people who are directly or indirectly employed by the biogas sector [18].

## 3. Biodiesel/Bio-Oil Production from Sewage Sludge

Several biofuels (bio-ethanol, bio-oil, bio-methanol, biodiesel, dimethyl ether, etc.) have been developed in recent decades, with biodiesel and bioethanol being the most commonly used [19]. Their acceptability is primarily attributed to their desirable qualities; requiring few or no adjustments to the present engine hardware and fuel supply configuration, having a more straightforward manufacturing procedure, and economic feasibility [44,45]. However, the use of bioethanol as an additive to gasoline is limited due to the required octane number, engine performance, and combustion quality. Thermal properties such as volatility, cetane number, and calorific value are comparable to commercial diesel; however, there are significant differences in fuel properties, such as viscosity, oxygen content, density, and stoichiometric fuel-to-air ratio, that must be taken into consideration. Biodiesel has a 10% lower heating value than conventional fossil diesel, but because of its higher specific gravity, biodiesel has only a 5% lower energy content per unit volume. As a result, the biodiesel-powered engine outperforms the standard diesel-powered engine in terms of thermal efficiency [46]. Furthermore, according to life cycle assessment statistics, biodiesel reduces GHG emissions with a net energy ratio greater than 1.30, whereas mineral

diesel has a net energy ratio of 0.19 [47]. Thus, biodiesel is a viable alternative to fossil fuels that is expected to reduce global environmental pollution and create jobs in rural areas [48].

Biodiesel is also known as "alkyl esters of fatty acid," and it is synthesized by reacting to various lipid fractions and alcohol in the presence of an enzyme, acid, or base catalyst [49,50]. Vegetable oils, the most common feedstock for biodiesel production, account for approximately 80% of total production costs and thus appear as the most significant barrier to the commercialization of biodiesel [51,52]. Furthermore, the production of biodiesel from vegetable oil conflicts with the need for human consumption. It is possible that waste cooking oils, which are less expensive, could be used as an alternative feedstock, but their varying properties affect the consistency of the biodiesel and necessitate additional purifying costs [49,52]. Researchers are currently looking into non-edible oil sources such as jatropha, algae, neem, or karanja as biodiesel feedstock alternatives; however, cultivation of these plants requires a large area of land, which is a major drawback [52–54]. Therefore, it is of the utmost importance to find a conveniently accessible and economical alternative for biodiesel feedstock.

Sewage sludge, being rich in organic content, is receiving global attention as a feedstock for biodiesel production. Biodiesel synthesis from sewage sludge is considered profitable due to its high lipid content [55–57]. Lipids are fatty acid-based organic compounds or their derivatives, such as free fatty acids (FFA), glycerides, phospholipids, and sphingolipids [58]. The presence of lipids in sewage sludge is a consequence of the absorption of these lipids from residential and industrial waste or due to the occurrence of cell lysis, during which plasma membrane breaks down and produces lipids. Figure 5 demonstrates the schematic process diagram of biodiesel extraction from raw sewage sludge. Numerous scientific studies carried out across the globe have reported that the lipids found in sewage sludge are a viable feedstock for biofuel generation, which, coupled with its economic implications on waste handling and management, is a promising option [59,60].

Boocock et al. [61] studied two main extraction methods to produce lipids from sewage sludge: the Soxhlet extraction method and the boiling extraction method. They used two solvents, toluene and chloroform, for the extraction process and found that chloroform was not suitable due to environmental concerns. They used 6 mL solvent/gm-dried sludge and found the yield of lipid 12 wt% (dried sludge basis) and 18 wt% (dried sludge basis) in the soxhlet extraction method and boiling extraction method, respectively. Table 1 presents the yield of lipids from sewage sludge using different solvents.

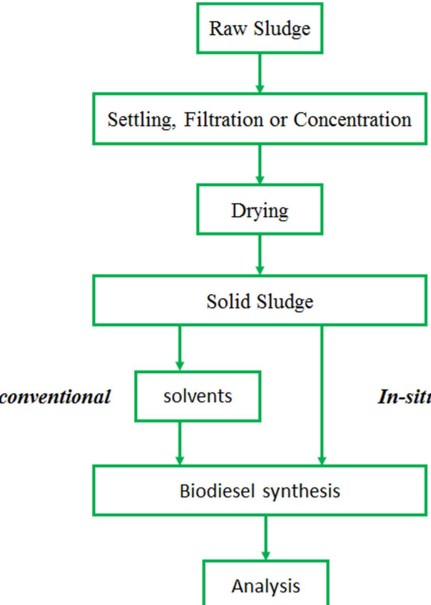

**Figure 5.** Flow diagram showing extraction process of biodiesel from raw sludge [62].

**Table 1.** Lipid yield from the sewage sludge using different solvents.

| Solvent | Lipid Yield in wt% (Based on Weight of Dried Sludge) | Reference |
|---|---|---|
| Hexane | 25.3 ± 1.3—primary sludge<br>9.3 ± 1.3—secondary sludge<br>21.9 ± 1.9—blended sludge (65% primary sludge and 35% secondary sludge)<br>10.1 ± 1.5—stabilized sludge | [59] |
| Hexane-Ethanol | 10.3 ± 0.2 | [63] |
| Hexane (non-acidic) | 27 ± 1—primary sludge<br>9 ± 1—secondary sludge<br>21 ± 1—blended sludge (65% primary sludge and 35% secondary sludge)<br>9 ± 1—stabilized sludge | [64] |
| Hexane (acidic) | 25 ± 1—primary sludge<br>7 ± 1—secondary sludge<br>20 ± 1—blended sludge (65% primary and 35% secondary)<br>10 ± 2—stabilized sludge | [64] |

### 3.1. Biodiesel Production Technology

Direct use and blending, microemulsion, transesterification, and thermal cracking (pyrolysis) are the four ways to synthesize biodiesel from the lipid-rich substrate [65]. Although the direct use and blending and micro-emulsion techniques are easy, they have numerous drawbacks, such as low fluidity product, decreased reactivity and volatility of unsaturated hydrocarbon, lubricating oil contamination, and carbon deposition in the engine [66,67]. The pyrolysis method generates biodiesel with low sulphur content, high fluidity, and high cetane number, but undesirable carbon content and ash [68]. Transesterification is the most widely used biodiesel synthesis technique since it is technologically simpler and produces a higher yield [69,70].

### 3.1.1. Biodiesel Production by Transesterification of Lipids Derived from Sewage Sludge

Dufreche et al. [71] first published a report on the generation of biodiesel from sewage sludge-derived lipids by ex-situ transesterification resulting in a biodiesel yield of 4.4 wt% (based on the total dry weight of activated sludge). In contrast, in-situ acid-catalyzed transesterification resulted in 6.23 wt% production. In acid-catalyzed in-situ transesterification, biodiesel is directly obtained from the lipids without the need of extraction processes.

Mondala et al. [51] found that at the reaction temperature of 75 °C and sludge-to-methanol (S/M) ratio of 1:12 by mass, a biodiesel production of around 14% for primary sludge and 2% for secondary sludge was obtained by in-situ transesterification utilizing a 5% $v/v$ $H_2SO_4$ catalyst.

Siddiquee & Rohani [72] studied the biodiesel synthesis of $H_2SO_4$-catalyzed transesterification and esterification of the lipid isolated from primary and secondary dried sludge. Based on lipid mass fraction, the production of esters from the primary sludge for methanol and hexane was around 40%. With the methanol as an organic solvent, the secondary sludge produced a maximum of 30% fatty acid methyl esters (FAME). The yield improved by roughly 18% ($w/w$) due to the inclusion of natural zeolite in the dehydration catalyst.

Olkiewicz et al. [59] improved and applied Christie & Han's [73] process for biodiesel production, yielding 13.9% (of sludge dry weight) from primary sludge and 10.2% from mixed sludge. The biodiesel yields from secondary and stabilized sludge were 2.9% and 1%, respectively. The high concentration of organic materials in primary sludge accounts for its high biodiesel output.

Pastore et al. [74] used dewatered sludge (15 wt% total suspended solids) as the raw material to eliminate the sludge drying phase. The 17 MJ/kg-FAME energy supply resulted in an 18% ester yield via a two-step process: First, lipid extraction from the $H_2SO_4$-acidified dewatered sludge, and then methanolysis. The use of $H_2SO_4$ is critical because it converts triglycerides into biodiesel and soaps into fatty acids via methanol esterification. The

primary sludge was valorized in order to produce valuable compounds such as aliphatic alcohols, wax, and sterols.

Zhang et al. [75] used a two-step process to produce biodiesel with a molar ratio of methanol to lipid of 6:1, $H_2SO_4$ as a catalyst at 1% $v/v$, and operating conditions of 50 °C. The researchers also tested a one-step biodiesel synthesis method in which dried sludge was immediately transesterified at 50 °C for 4 h using $H_2SO_4$ as a catalyst. For producing one ton of biodiesel, the two-step procedure with a total applied energy of 12.4 GJ was found more effective and practical than the one-step procedure with an applied energy of 17.3 GJ.

Choi et al. [76] described an alternative method for producing biodiesel from wet sludge using xylene instead of hexane as the solvent and $H_2SO_4$ as the acid catalyst for in-situ transesterification. The biodiesel yield was reported to be 8.12%, which was 2.5 times higher than the yield produced by using hexane.

Melero et al. [77] used Zr-SBA-15 as a catalyst at 209 °C, 2000 rpm, and 12.5 wt% (lipid mass basis) loading of the catalyst. Under identical reaction conditions of 65 °C and 3 h, the one-step approach converted saponifiable fatty acids and glycerides more effectively than the two-step process. The highest ester output was measured using primary sludge, and the single-step technique was around 15%.

Tran-Nguyen et al. [78] investigated sludge transesterification using a subcritical acetic acid and methanol mixture (5% acetic acid and 85% methanol). Keeping the temperature at 250 °C and the solvent-to-sludge ratio at 5 (mL/g) resulted in a FAME yield of 30.11% in 30 min. When acid-catalyzed ($H_2SO_4$ 4% $v/v$) transesterification was used, a higher yield of 35% was obtained, but the process was too long, lasting 24 h at 55 °C and producing a methanol to sludge ratio of 25 (mL/g).

Urrutia et al. [79] synthesized biodiesel from oily and secondary sludge using enzymes (Novozym 435) and $H_2SO_4$ as a catalyst. Triacylglycerols (TAG) and free fatty acids (FFA) were converted to FAME by the two catalysts. The esterification of FFA using acid catalysis outperformed TAG transesterification. According to the observations, the enzyme-based catalyst hydrolyzed the TAG into FFA before esterifying it into methyl esters. FAME yield was approximately 63% for oily sludge treated with an acid catalyst and 58% for enzyme as a catalyst.

Demirbas et al. [80] found a 24% yield of methyl esters from the lipids extracted in a similar study [71,79] to convert sewage sludge to biodiesel. The average amount of saturated fatty acids collected was 55.0%, with 37.5% palmitic acid being the most abundant and 12.0% stearic acid as a second majority. Linoleic acid (6.2%) and oleic acid (29.0%) made up the majority of the unsaturated fatty acids collected.

Choi et al. [81] developed a biorefinery method for manufacturing biogas and FAME from the waste activated sludge (WAS) through in-situ extraction of lipids and AD. The recorded yield of the non-polar trans-esterified product was found to be 9.68% (WAS dried mass based), and the yield of FAME was recorded at 78.45% (saturated content-40.7% and unsaturated content-59.3%) at the optimal reaction conditions of 2.5 methanol/hexane ratio by volume, 10 mL/g methanol dose, 55 °C reaction temperature, and 8 h of reaction time.

Patiño et al. [82] used acid-catalyzed in-situ transesterification to convert sludge-extracted lipids to biodiesel. It was noted that the application of the ultrasonication step as sludge pre-treatment significantly increased (over 65%) the yield of FAME for both the in-situ and acid transesterification with only 16 h of contact time. In contrast, without ultrasonication pre-treatment, approximately 24 h of contact time was required. After 50 min of ultrasonication, a higher degree of cell destruction was achieved, and a massive amount of accessible lipid was exposed to the catalyst and solvent.

Various studies on production of biodiesel through transesterification of lipids derived from sewage sludge are summarized and compared in Table 2.

**Table 2.** Comparison of biodiesel production by transesterification of lipids derived from sewage sludge.

| Lipid Extract from Sewage Sludge in wt% (Dried Sludge Basis) | Production Technology | Methanol to Sludge/Lipid Mass Ratio (mL/g) | Fame Yield in wt% (Dried Sludge Basis | Reference |
|---|---|---|---|---|
| 27.43 | In situ transesterification, $H_2SO_4$ 1% *v/v*, 50 °C, 24 h | 5 | 6.23 | [71] |
| - | In situ transesterification, $H_2SO_4$ 5% *v/v*, 75 °C, 8 h | 12 | 14.5 (primary sludge) 2.5 (secondary sludge | [51] |
| - | In situ transesterification, $H_2SO_4$ 4% *v/v*, 55 °C, 24 h | 30 | 4.79 | [83] |
| Methanol extraction 14.46 (primary sludge) 10.04 (secondary sludge) Hexane extraction 11.06 (primary sludge) 3.04 (secondary sludge) | Transesterification, $H_2SO_4$ 2% *v/v*, 60 °C, 24 h | 50 | Methanol extraction 26.84 (primary sludge) 30.28 (secondary sludge) Hexane extraction 41.25 (primary sludge) 38.94 (secondary sludge) (on the basis of lipid) | [72] |
| - | In situ transesterification, $H_2SO_4$ 10% *v/v*, 75 °C, 24 h | 30 | 3.93 | [52] |
| 66.64 | Subcritical methanol-subcritical water, 175 °C, 8 h, 3.5 MPa | 30 | 45.58 | [84] |
| 25.3 (primary sludge) 21.9 (blended sludge) 10.1 (stabilized sludge) 9.1 (secondary sludge) | Transesterification, $H_2SO_4$ 1% *v/v*, 50 °C, 24 h | 20 | 13.93 (primary sludge) 10.9 (blended sludge) 2.9 (stabilized sludge) 1 (secondary sludge) | [59] |
| - | $H_2SO_4$ 0.05% *v/v*, n-hexane as co-solvent, 55 °C, 8 h | 10 | 9.68 | [76] |
| 44.40 | $H_2SO_4$ 7 wt%, n-hexane as co-solvent 10% *v/v*, 55 °C, 7 h | 20 | 61 | [85] |
| 45.42 | Acetic acid—subcritical methanol (AA- 15%) 250 °C, 1.5 h, 2.5 Mpa | 5 | 30.11 | [78] |
| 65.4 (greasy sludge) | In situ transesterification, ($H_2SO_4$ and Novozym 435), 60 °C, 8 h for acid catalyst, 16 h for enzyme catalyst | 15 for acid catalyst 4 for enzyme catalyst | 63.9 (acid catalyst) 58.7 (enzyme catalyst) | [79] |
| 14.5 (sonicated sludge) | Two-step biodiesel production process, 0.2% (g/g-lipid) $H_2SO_4$, 70 °C, 8 h | 20 (lipid basis) | 5.3 | [86] |

### 3.1.2. Biodiesel Production from Sewage Sludge by Cultivated Microorganisms

Prior to converting the sewage sludge into methane and carbon dioxide, aerobic microbes can be used to extract lipids from the carbon moieties in the sludge. According to a study, yeasts can only accumulate up to 70% of the lipids in dry matter [87]. The first data on lipid storage and fermentation conditions were provided by Mulder et al. [88]. Microbes require a high C/N ratio to extract the lipids effectively. Among the yeasts studied, large amounts of lipid storage were reported in lipomyces starkeyi when cultivated on sludge media [89]. Oleaginous microbes can concentrate more than 20% (g/g) of lipids [90]. Among the 60,000 known fungal species, only about 50 of them are known to accumulate a lipid content greater than 25% [90,91]. Because of their rapid lipid accumulation and ability to thrive in a variety of industrial and agricultural waste conditions, oleaginous microorganisms are considered well suited for the purpose [92]. Furthermore, unlike vegetable oil, oleaginous microorganism culture is unaffected by changes in climatic conditions.

Selvakumar & Sivashanmugam [93] studied the impact of thermal, chemical, and sonication-based pre-digestion of the municipal WAS for oleaginous L. starkeyi MTCC-1400 cultivation as a model organism for the generation of high-yield lipid and biomass. When WAS was treated with NaOH and KOH at an energy input of 5569 kJ/kg, suspended solids (SS) were reduced, and COD was solubilized. The NaOH-pre-digested sample had the highest fat concentration of 64.3% and the highest biomass concentration of 17.52 g/L. The majority of lipid profiles contain 38.2% and 45.6% of oleic acid and palmitic acid, respectively.

The primary conclusion drawn from the research is that acidified trans-esterification is more effective than the other three methods, with the most limiting step being lipid extraction. As a result, some researchers proposed the application of ultrasonication technology as an appropriate pre-treatment to supplement lipid extraction. High-magnitude ultrasonic irradiation in the sewage sludge facilitated cell lysis, resulting in the release of the lipids into the medium.

The physical and chemical impacts of a heterogeneous reaction process are amplified by the rapid development and collapse of cavitation bubbles caused by sonication. The radial motion of the bubbles creates microturbulence, emulsifying the reactant and producing the physical effect. At the same time, free radicals released during bubble collapse account for the chemical effect. The large interfacial area acts as a mechanical agitator, while free radicals in the bulk reaction mixture accelerate the reaction rate [94]. On account of all these phenomena taking place, ultrasonic irradiation could be used as a novel and effective integration technology in biodiesel production [95]. This method has been successfully used in recent times to produce biodiesel and biogas from algae and sewage sludge as feedstock [96,97].

### 3.1.3. Bio-Oil Production from Sewage Sludge Using Pyrolysis

Pyrolysis is a thermo-chemical process that occurs at elevated temperatures in the absence of oxygen. It results in the breakdown of large molecular chemical compounds into smaller molecules and yields oil, gaseous, and solid fractions. Pyrolysis oil, also known as bio-oil, is produced by heating dry biomass in moderate to high-temperature reactors at ambient pressure without using a solvent [65]. It can also be produced by subjecting the sludge to hydrothermal liquefaction using solvents under high-pressure and low-temperature conditions [98]. Many organic components, such as acids, alcohols, ketones, and oligomers, are found in bio-oil [99]. Bio-oil is a renewable and sustainable fuel with properties comparable to conventional diesel. Viscosity, high acidity, corrosiveness, instability, and low calorific value are some major issues preventing its commercialization [100,101]. Making bio-oil from sewage sludge is advantageous due to it being a low-cost feedstock with steady availability. However, sludge-based bio-oil has a substantial amount of N and O [102], which prevents its commercial use due to the aforementioned negative fuel qualities [103]. Without modification, some of its properties limit its utility as an alternative transportation fuel [104]. It could be used in the production of some raw materials such as aromatics, olefins, hydrogen, and chemical extraction, as well as used directly in boilers and furnaces [105]. Bio-oil, on the other hand, could be improved by blending it with biodiesel to increase its combustibility [98].

Capodaglio et al. [106] used microwave pyrolysis (2.45 GHz, 3 kW) to generate sewage sludge pyrolysis oils (SSPOs) from pre-digested sewage sludge. Although sludge-produced oil has low "lower calorific value" (LCV) as compared to conventional biodiesel made from vegetable oil, the rest of the fuel properties are comparable. The LCV of recovered oil, 32.976 kJ/g, is within the required range of 32–34 kJ/g, in contrast to maize and safflower oils, which have LCVs of 42–43 kJ/g. As a result, the LCV of oil produced by sludge have 30% less value than conventional diesel.

Alvarez et al. [105] designed a conical spouted bed reactor with a temperature range of 450–600 °C for sewage sludge pyrolysis. The scientists identified amides, alcohols, ketones, nitriles, and phenols as the most abundant organic components in the bio-oil produced at 500 °C (highest yield at this temperature; 77% on an ash-free dry weight basis). Because bio-oil contains fewer oxygenated molecules than other lignocellulosic materials, it has a higher carbon and hydrogen concentration and better "lower heating value" (LHV).

Arazo et al. [104] used the nickel-based catalyst HZSM-5 to produce bio-oil from the sewage sludge. The effects of different reaction conditions on the production of bio-oil and degree of deoxygenation and denitrogenation were investigated. With increased process temperature, the biodiesel output was found to drop while the oxygen and nitrogen removals were enhanced. Response surface methodology (RSM) revealed that the optimal

set of process parameters for producing bio-oil with a 67.2% yield included a temperature of 258.5 °C, a mass ratio of 2.50 ethanol to bio-oil, and a reaction duration of 3.23 h.

### 3.1.4. Bio-Oil Production from Sewage Sludge by Supercritical Fluid Conditions

Leng et al. [98] used sludge from various sources, including sewage sludge, to produce bio-oil by supercritical liquefaction at 300 °C with methanol as the solvent. This method yielded bio-oil that was 20–30% C-rich and viscous, with a high concentration of methyl esters. The extracted bio-oil was modified by combining it with FAME, either by micro-emulsifying or blending. According to the findings of the studies, the process of blending or micro-emulsification increased the breakdown rate of hybrid fuel systems while decreasing their activation energy requirements. Furthermore, the enhanced ignition characteristics of the blended micro-emulsions make it better suited for stand-alone use bio-oil or biodiesel, which otherwise has critical disadvantages such as higher viscosity and longer ignition delay.

The effect of mixed solvents such as n-hexane-water (HEX-$H_2O$) and methanol-water (MeOH-$H_2O$) on the extraction of wet sewage sludge was investigated [107]. Using MeOH-$H_2O$ with 37.4% methyl esters, the study reported a maximum yield of 46.5 wt% and a minimum yield of 10.3 wt% using HEX-$H_2O$. HEX-$H_2O$ generated bio-oil with the highest calorific value ($36.45 \pm 0.38$ MJ/kg) and the lowest oxygen content ($7.65 \pm 0.26$ wt%). The use of MeOH-$H_2O$ promoted the formation of esters, whereas HEX-$H_2O$ not only improved the extraction of aliphatic compounds but also enhanced the thermal stability of the residue.

The primary barriers to bio-oil commercialization are its low yield and low fuel quality. Due to the high operating costs resulting from high temperature requirements for bio-oil synthesis, it is primarily not recommended for industrial-scale production. To overcome this, a catalyst could be used to synthesize it at a lower temperature with improved fuel quality, without deploying any other upgradation techniques like emulsification. For a better understanding of bio-oil yield, it is recommended to understand the catalytic mechanism that occurs inside the pores of the catalyst. Kinetic studies can be carried out to better predict the thermal behavior of sludge lipids during catalytic pyrolysis.

### 3.2. Biodiesel Quality

Biodiesel intended for commercial use must meet the standard specifications laid out by the concerned authorities. Two such specifications, ASTM D-6751 and EN-14214, are shown in Tables 3 and 4, specifying biodiesel standards to be used in the engines without requiring hardware modifications. It has been a major challenge for researchers to ensure the qualitative consistency of biodiesel derived from the lipids extracted from sewage sludge. Sludge comprises different lipid sources like fatty acids, triglycerides, bacterial lipids, and phospholipids, in addition to chemicals like steroids, hydrocarbons, terpenoids, wax esters, benzene, polyhydroxy-alkanoates, aromatic hydrocarbons, and other pharmaceutical compounds [108,109]. Apart from lipids extraction, there is ample opportunity to extract different chemical compounds, which may affect the gravimetric yield of biodiesel. The cracking method can potentially transform these species and significantly increase the yield of FAME from sewage sludge [83,110].

**Table 3.** ASTM D-6751 biodiesel standard specifications.

| Parameters | Unit | Method | Limit |
|---|---|---|---|
| Acid number (max) | mg KOH/g | D 664 | 0.5 |
| Carbon residue (max) | % by wt | D 4530 | 0.05 |
| Cetane number (min) | - | D 75 | 47 |
| Cloud point | °C | D 2500 | −3 to 12 |
| Copper strip corrosion at 50 °C (max) | - | D 130 | 3 |
| Flash point (min) | °C | D 93 | 93 |
| Oxidation stability (min) | hours | EN 14112 | 3 |
| Phosphorus (max) | % by wt | D 4591 | 0.001 |
| Pour point | °C | D 97–05 | −15 to −16 |
| Sulphated ash (max) | % by wt | D 874 | 0.02 |
| Total glycerine (max) | % by wt | D 6584 | 0.24 |
| Total sulphur (max) | % by wt | D 5453 | 0.05 |
| Viscosity (Kinematic) | $mm^2$/s | D 445 | 1.9 to 6 |

**Table 4.** EN-14214 biodiesel standard specifications.

| Parameters | Unit | Method | Limit |
|---|---|---|---|
| Acid number (max) | mg KOH/g | EN 14104 | 0.5 |
| Carbon residue (max) | % by wt | EN ISO 10370 | 0.3 |
| Cetane number (min) | - | EN ISO 5165 | 51 |
| Density at 15 °C | $kg/m^3$ | EN ISO 3675/12185 | 860 to 900 |
| Diglycerides | % by wt | EN 14105 | 0.2 |
| Flash point (min) | °C | EN ISO 3679 | 101 |
| Free glycerine (max) | % by wt | EN 14105 | 0.02 |
| Monoglycerides | % by wt | EN 14105 | 0.8 |
| Oxidation stability (min) | hours | EN 14112 | 8 |
| Phosphorus (max) | % by wt | EN 14107 | 0.0004 |
| Sulphated ash (max) | % by wt | EN ISO 3987 | 0.02 |
| Total contamination | % by wt | EN 12662 | 0.0024 |
| Total glycerine (max) | % by wt | EN 14105 | 0.25 |
| Total sulphur (max) | % by wt | EN ISO 20846/20884 | 0.001 |
| Triglycerides | % by wt | EN 14105 | 0.2 |
| Viscosity (Kinematic) | $mm^2$/s | EN ISO 3104 | 3.5 to 5 |

*3.3. Process Economy*

Biodiesel derived from renewable sources has several environmental benefits, making it a viable alternative fuel. Indeed, the product's high price and limited availability are the primary barriers to commercialization. The expense of raw material procurement and processing, in particular, contributes to the high production price of biodiesel, which limits its commercial expansion compared to traditional diesel [83,111].

Sewage sludge, a waste that must be cleaned before disposal, dramatically reduces the cost of biofuel production when used as a raw material. Biodiesel may be produced by extracting lipids from sewage sludge using organic solvents. The recovery of these solvents over 99% can be achieved [56]. Using mixed sludge as feedstock, the projected production cost of biodiesel in 2010 was reported to be 3.11 to 3.23 USD/gallon, compared to 4.00 to 4.50 USD/gallon for biodiesel derived from processed soy oil and 3.00 USD/gallon for conventional fossil-derived diesel [51,56].

According to a cost analysis study [56], several solvents, such as ethanol, methanol, hexane, and toluene, were utilized for lipid extraction. The study assumed that the raw material cost was not too high, while the drying cost was accounted for the analysis. Toluene and hexane were the least expensive of the four solvents, costing 2.79 and 2.89 USD/gallon, respectively. Even though toluene is less costly, due to its high boiling point its consumption was more.

The breakeven selling cost of biofuel (0.49 USD/l) as calculated by Xin et al. [112] was similar to the figure obtained by Orfield et al. [113]. Xin et al. [112] estimated it costs at less than 0.55 USD/l to synthesize bio-oil from algae, comparable to the 80 USD/barrel cost of producing conventional fossil fuel oil. Xin et al. [114] determined the integrated system to be superior to the segregated system, despite this estimate being lower than the biofuel output from algae derived from centrates, which cost 0.59 USD/l. According to Di Fraia et al. [115], the simple payback time of the co-generation system for sewage sludge treatment is 6.8 years. According to the research, the simple payback time of an integrated wastewater biofuel system is about 9.05 years, 8.08 years, and 7.47 years at crude oil equivalent prices of 74.14 USD, 80.43 USD, and 85.15 USD, respectively. The estimated cost of oil production by pyrolysis of sludge was 6.9 €/kg-ds (digested sludge), which was in perfect accord with the results reported by a previous study [116]. From an economic analysis, using sewage sludge for biodiesel/bio-oil production could reduce environmental burdens and potentially become profitable. Table 5 summarizes the many obstacles encountered on the way to biodiesel manufacturing.

**Table 5.** Major challenges associated with commercialization of biodiesel production [45].

| S.No. | Major Challenges | Description |
|---|---|---|
| 1 | Sludge collection | Collection of sludge from different sources, followed by pre-treatment |
| 2 | Sludge drying | Drying large quantities of sludge requires high capital and infrastructure investment |
| 3 | Lipid extraction | Requirement of huge volumes of expensive solvents makes the process very expensive |
| 4 | Biological processing | Removal of metallic compounds and pharmaceutical from sludge |
| 5 | Extraction and purification of bio-oil | Simultaneous extraction and acid transesterification makes the production more efficient; soap formation as process by-product hinders the extraction of biodiesel and glycerol; application of advance techniques like ultrasound and microwave improves the yield |
| 6 | Quality enhancement | Final product must comply with ASTM standards |
| 7 | Reactor design | Efficient production of bio-oil largely depends upon optimum design and operation of the bioreactors |
| 8 | Economic consideration | The net cost of biodiesel production must be considerably lower than the conventional fuels (petrol and diesel) |

## 4. Sewage Sludge as a Source of Heavy Metal

Over the past few decades, it has been observed that incoming sewage at different WWTPs has significant quantities of heavy metals due to the intrusion of various industrial and other wastes [117]. Wastewater treatment operations result in the accumulation of these heavy metals in the sewage sludge [118]. The heavy metal resource potential of the sewage sludge is promising, but its recovery is still a huge challenge due to the process being uneconomical [25]. It is crucial to highlight that the sewage sludge components were evaluated at 480 USD/dry ton (compared to 7 USD/dry ton for phosphorus), with Cu, Fe, Zn, Pd, Ti, Ir, Cr, Ga, Mn, Au, Cd, Al and Ag having the highest potential for economic recovery [26]. The majority of metals are deemed harmful to the ecosystem, and thus their presence restricts the sludge to be used for land application [119]. Thus, recovery of the metals from the sewage sludge in an efficient way could be a huge gain for both the economy and the environment. Bioleaching is a promising technique for heavy metal recovery from the raw and digested sewage sludge due to the ability of certain microorganisms to extract metals from polluted solid matrices. Besides this, there are some other oxidative bioleaching techniques, but problems such as low removal efficacy and high costs of aeration restrict their application.

*Novel Enhanced Anaerobic Bioleaching Process for Metal Recovery from Sewage Sludge*

The suggested innovative technique is an improved anaerobic bioleaching process which could be a promising way to leach metals from sewage sludge via the metal's recovery stage from neutralized, intermediates-rich supernatant while also contemporarily improving AD to recover bio-methane and produce a final "clean" stabilized digestate.

Bio-leaching removes heavy metals from the sludge through chemical redox, biological oxidation, complexation, adsorption, and dissolution, among other processes [120]. As shown in Figure 6, the sludge, after appropriate pre-treatment (thermal hydrolysis or ultrasonication), is subjected to the acidogenic phase to enhance hydrolysis, resulting in the leaching of the metals bound in the solid phase into the liquid phase. An appropriate solid-liquid separation technology (such as centrifugation or membrane filtration) is used to separate the metals and volatile fatty acids (VFA) rich supernatant from the solid phase. Metals can be recovered by selective adsorption/precipitation from the supernatant. At the same time, the recycled VFA-rich supernatant coupled with the solid fraction is subjected to AD, resulting in improved bio-methane production and a good quality digestate that could be utilized as a bio-fertilizer.

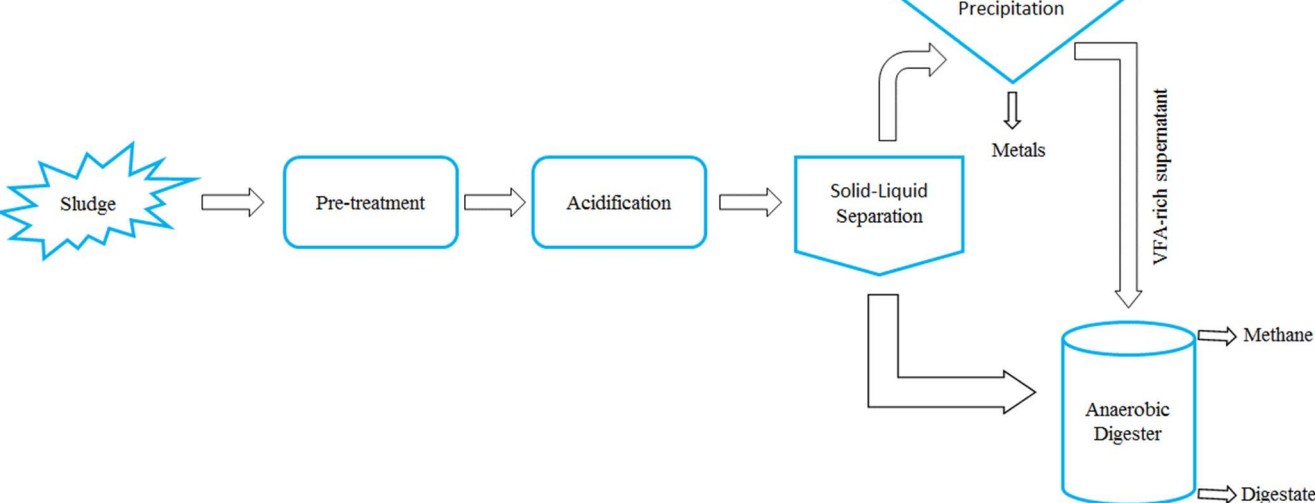

**Figure 6.** Integrated metal recovery scheme from sludge [26].

A study carried out at the Water Research Institute found that solids disintegration and organics solubilization with the application of sonically-aided thermal hydrolysis improved metals release. In particular, the statistically significant positive association between organics disintegration degree and the concentration of soluble Co, Se, As, Zn, Ni, Ga, and V revealed that the organically-bound fraction constitutes a significant proportion of these elements.

During AD of sludge, the conversion of metals from one form to another is mainly associated with the reduction of metal-bound organics. Figure 7 compares the partitioning coefficients of the elements between the particulate and liquid phases in WAS and full-scale anaerobic digestate to identify which metals are mobilized during the whole digestion process. While Fe, Pb, Al, and Mn tend to collect in particulates, Cu, Ni, As, V, Se, Zn, Ga, and Cr tend to dissolve, enhancing their mobility and availability.

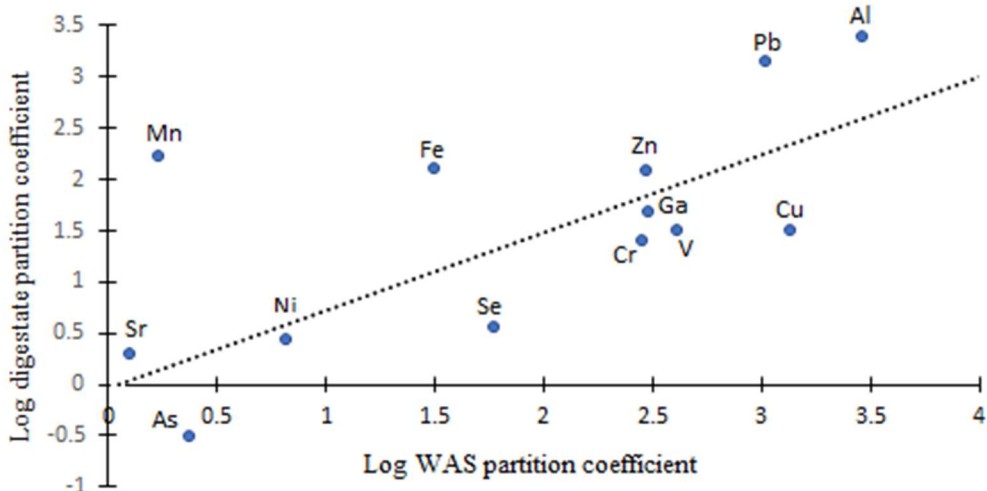

**Figure 7.** Distribution of metal during full-scale anaerobic digestion [26].

Using anaerobic bacteria and acidification to enhance metal recovery from sludge without any chemical leaching agent or external substrate is a promising and eco-friendly approach. This bioleaching-based metal separation process from the sludge also enhances the production of a clean energy source in the form of biogas.

## 5. Sewage Sludge as a Source of Nutrient/Fertilizer

Being rich in nutrients, sewage sludge has enormous utility in terms of nutrient recovery and reuse. It is notably rich in P, an essential element for plant development, which occurs in limited quantities in nature. When these nutrients are dumped into water bodies, they cause nutrient pollution or eutrophication, which has negative impacts on the quality of the water as these nutrients promote the overgrowth of algae in the receiving water bodies. Table 6 shows the typical nutritional composition of digested sludge. Though rich in nutrients, sewage sludge is also characterized by the presence of heavy metals that have negative impacts on soil properties, and they may also result in bioaccumulation through the food chain. Thus, it is necessary to have regulations on the heavy metal content of the sewage sludge to be used as fertilizer. Table 7 presents permissible limits of heavy metal concentration in the sewage sludge to be used for land application.

**Table 6.** Typical nutrient profile of sludge digestate [18].

| Nutrient | Concentration (kg/t-Fresh Weight) |
|---|---|
| Total Nitrogen (N) | 11 |
| Total Phosphate ($P_2O_5$) | 5.5 |
| Total Potash ($K_2O$) | 0.5 |
| Total Magnesium (MgO) | 1.6 |
| Total Sulphur ($SO_3$) | 8.2 |

**Table 7.** Permissible limits of heavy metal concentration in sludge for land application [24].

| Heavy Metal | Permissible Limits (mg/kg-Dry Weight) | |
|---|---|---|
| | USEPA 40 CFR Part 503 | EU Commission 86/278/EEC |
| Cu | 1500 | 1000–1750 |
| Zn | 2800 | 2500–4000 |
| Ni | 420 | 300–400 |
| Cd | 39 | 20–40 |
| Pb | 300 | 750–1200 |

*AirPrex® Patented Sludge's Nutrient Recovery Technology*

AirPrex® technology involves aerating the digestate in a specialized reactor to equilibrate alkalinity to atmospheric conditions. After that, magnesium is introduced into the reactor, where it precipitates with the soluble phosphorus to form struvite [121]. The resulting change in digestate chemistry results in a decrease in polymer consumption and an increase in dewatered biosolids concentration. It has been claimed that the effect of cations on biosolids' dewaterability helps explain the observed increase in the dewatering performance of Bio-P-affected biosolids [122]. One of the prominent theories is the hypothesis of divalent cation bridges (DCB). This theory holds that divalent soluble cations such as calcium ($Ca^{2+}$) and magnesium ($Mg^{2+}$) bridge negatively charged binding sites on extracellular polymeric substances (EPS), thus stabilizing biosolids and improving bio flocculation [123–125].

AirPrex® reactors are deployed to produce struvite solids from soluble phosphate ($PO_4^{3-}$) entering the reactor. As indicated in Figure 8, the produced struvite is divided among three flow channels. A portion of the total struvite solids produced in the AirPrex® reactor settles and exits with the underflow as the recovered product (fp). The remainder stays inside the biosolids matrix and leaves the reactor through the overflow. The biosolids are transferred from the reactor to the dewatering process, where the struvite is further divided into the dewatered biosolids (fb) and the dewatering liquor (centrate) (fc). It is crucial to establish the quantity of struvite generated and its partitioning. Since the struvite retained in the dewatered biosolids increases the dry mass that requires transport and land application, the struvite in the centrate fraction dictates the plant recycling phosphorus load, while the recovered fraction dictates the overall product mass. From the dissolved and total $PO_4^{3-}$ and $Mg^{2+}$ concentrations, the struvite proportion in the centrate can be calculated directly. Due to the pilot arrangement, it was not feasible to determine the recovered product fraction directly. To determine the amount of additional struvite retained in the biosolids stream (fb), acid extraction assays were conducted on untreated and treated biosolids to determine the amount of $Mg^{2+}$ and $PO_4^{3-}$ released from the cake sample following acidification. This test gave an estimate of the proportion of the total dewatered biosolids mass attributable to the struvite for each solid stream, which could be used in conjunction with the known struvite production to calculate fb. A case study [126] of a pilot-scale AirPrex® reactor is presented and discussed below.

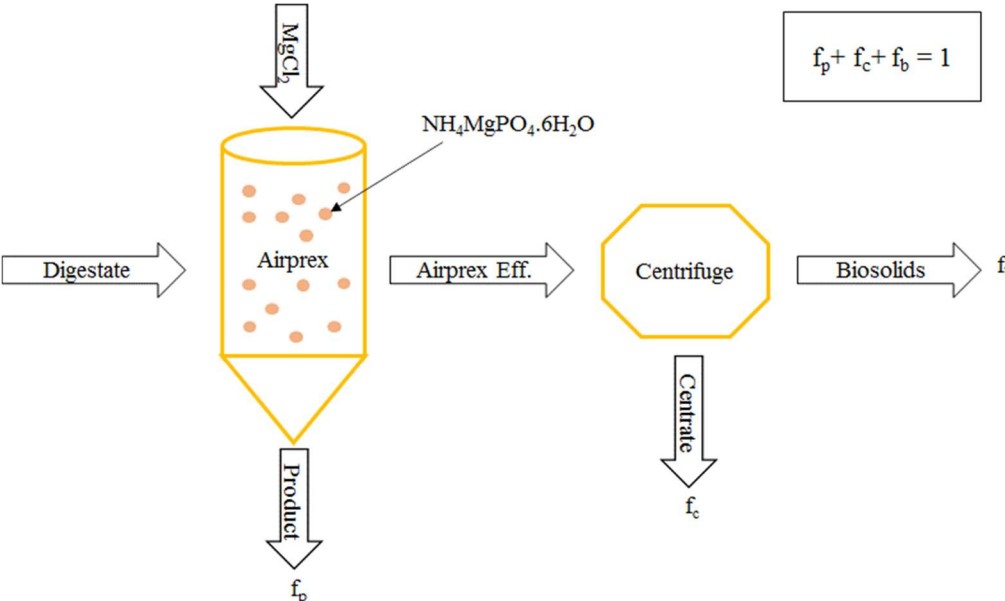

**Figure 8.** Process flow diagram showing struvite formation and partitioning in AirPrex® [126].

The 833 MLD Robert W. Hite Treatment Facility of the Metro Wastewater Reclamation District catered to the requirement of 1.8 million people. This facility with AD was required to integrate enhanced biological phosphorus removal (EBPR) to combat the excessive enrichment of nutrients in the South Platte River basin. Due to the well-established undesirable operational implications of Bio-P, including adverse effects on anaerobic solids digestion and dewatering processes [127,128], the district was investigating different phosphorus recovery systems. The district undertook a comprehensive pilot-scale assessment of AirPrex® as a potential method for mitigating the negative impacts of P on sludge treatment and handling procedures.

A 40-foot-tall AirPrex® system with provisions for struvite recovery, $MgCl_2$ feed, and a 10-inch diameter centrifuge for dewatering comprised the primary apparatus for the pilot-scale study. Analysis was carried out for the inflow of 41.6 lpm. Two sparging rings mounted at different heights inside the reactor received air from a turbo blower and an air compressor. Air was sparged into a reactor to mix its contents and to remove $CO_2$ to raise the pH to around 7.8. As Mg concentration is often the limiting factor in struvite precipitation, a 30% $MgCl_2$ solution was added to the digestate reactor to achieve the required Mg:P molar ratio. The molar feed ratio of magnesium to soluble phosphorus (Mg:P) ranged from 1.1:1 to 1.5:1. After precipitation, the struvite crystals (about 1.7 specific gravity) grew larger, with a portion settling in the conical part of the reactor. The non-settled struvite crystal fraction remained inside the biosolids matrix and was discarded with the reactor effluent. Settled struvite was collected from the reactor periodically via a discharge valve at the bottom of the reactor with the flow channel to a grit classifier. The struvite was purified with plant water by the classifier. The classifier released the gathered struvite into a collecting container.

The pilot centrifuge was designed to intake raw digestate from the holding tank or directly from the AirPrex® reactor. The centrifuge was operational 6 to 8 h a day, Monday through Friday, to develop dewatering efficiency plots across a fixed range of polymer dosing for both untreated and AirPrex®-treated digestate. Initial tests were done using two different commercially available polymers (Praestol K260-FL and K290-FL); however, after dose optimization, the bulk of the tests were conducted with BASF Zetag 8849 polymer. Results of the study revealed that pre-treatment led to a 15 to 20% decrease in polymer requirement for dewatering and a 2 to 3.5% rise in dry dewatered biosolids concentration, resulting in an overall 7 to 10% reduction in facility hauling needs.

## 6. Perspective and Concluding Remarks

Management of gigantic quantities of sewage sludge produced in present times poses many environmental challenges. Handling of sludge to comply with today's standards produces many economic liabilities in the form of infrastructure installation and operation costs. Hence, in recent times the focus on sludge management has been on the development of sustainable techniques that could harness the potential of sewage sludge streams as a source of energy and material. This is achieved by subjecting the sewage sludge to specific pre-treatment techniques before undergoing conventional treatments like AD. CAMBI® and BIOTHELYS® have successfully developed a thermal-hydrolysis-based pre-treatment technique to improve the digestibility and dewaterability of the sludge, significantly enhancing the biogas production from AD. AnDMBR technology has achieved similar goals by combining AD with a dynamic membrane for solid-liquid separation.

Biofuels, a potential alternative to fossil fuels, are derived by mixing high lipid feedstock with organic solvents. The use of vegetable oils, the most conventional feedstock for bio-oil production, make this process economically unsustainable. Sewage sludge, being rich in lipid content, could serve as a continuous and sustainable feedstock for biofuel production. Transesterification and thermal cracking are the most commonly used techniques for biofuel production from sewage sludge. Despite the abundant availability of sewage sludge, several factors limit the commercialization of biofuel production. The major factors include high sludge pre-treatment costs, the requirement of large volumes

of organic solvents for lipid extraction, and quality control of the product to comply with standard specifications.

Techniques have also been developed to recover certain heavy metals and nutrients from sewage sludge. A newer technique to recover metals involves pre-treating the sludge, followed by acidification to enhance hydrolysis, resulting in the leaching of the metals from the solid phase into the liquid phase. This is a novel method based on bioleaching to extract metals from the sludge without using any chemicals. Sewage sludge also contains substantial fractions of nutrients, such as N, P, S, etc., that can be recovered economically. AirPrex® technology facilitates the recovery of P from the digestate via the formation of struvite.

This resource recovery from sewage sludge is beneficial in various ways. Firstly, the extracted resource itself has its own economic value, driving the whole extraction approach's commercially viability. Secondly, these recovered resources could potentially fulfil the demand of certain quantities of their conventional counterparts, hence reducing the carbon footprints associated with the primary or secondary production of that resource. Thirdly, pre-treatment of sewage sludge for resource recovery also results in generation of better quality digestate after AD, which is relatively easier to manage.

In conclusion, recent advancements in sustainable techniques for energy and material recovery from sewage sludge have expanded the options available for treating and utilizing valuable resources. These advances have made it possible to recover more energy and materials from sewage sludge, leading to greater efficiency and lower environmental impact. Commercial-scale projects have demonstrated the viability of these techniques, and their continued development and implementation is expected to bring additional benefits in the near future. However, it is important to consider the social and economic aspects of these technologies to ensure that they are accessible, affordable, and suitable for different communities. To summarize, recent advancements in sustainable techniques for energy and material recovery from sewage sludge are a positive step towards a more sustainable future and efficient management of waste.

**Author Contributions:** Writing—original draft preparation, M.I.S. and H.R.; writing—review and editing, I.H.F. and F.B. All authors have read and agreed to the published version of the manuscript.

**Funding:** This research received no external funding.

**Data Availability Statement:** Not Applicable.

**Acknowledgments:** The authors would like to acknowledge Zakir Hussain College of Engineering and Technology, Aligarh Muslim University, for providing administrative and technical support.

**Conflicts of Interest:** The authors declare no conflict of interest.

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
