# Peer review of "Recent Advancement in Commercial and Other Sustainable Techniques for Energy and Material Recovery from Sewage Sludge"

_water, doi:10.3390/w15050948_

Round 1

Reviewer 1 Report

This work provides a wide review of commercially developed and still under research technologies for energy and other resource recovery of sewage sludge, while addressing the techniques, limitations and potential ways to improve their yields. However, there are some issues should be addressed in this work;

1) At lines 44-46, the sentence, "One of the studies..is 70 g/day". It is not quite clear what do authors want to express here. 

2) Some expressions could be polished or make it more straightforward, such as at lines 128-129, "Increasing amounts of xxx...otherwise be lost.." at lines 781-782..Since almost 61% ... implying 39% of ..." Authors are suggested to read through the whole manuscript to polish their writing. 

3) Authors have reviewed and listed a lot of available technologies regarding the treatment of sewage sludge, in this work. However, I did not see their own understandings about these technologies. I would advide authors to comment promising technologies in terms of limitations and techniques to break through the challenges. 

4) In addition to simply introduce the flow chart of some technologies, such as in Figure 3, Figure 4, Figure 6...It will be more useful to explain the fundamental mechanisms of these technologies using some simple words. 

5) Figure 5 dose not seem very useful. Can authors provide more useful information in this Figure?

6) The economical discussion section is important in this manuscript. Can authors provide a table or figure to demonstrate this part information more clearly. 

7) At line 457, the energy input is as high as 5569 kJ/kg, which is more than double of the vaporization enthalpy of water. Such high energy input can still have economical value?

8) some typos: at line 41, the unit of m3/year. 

      at line 448, a dash line should be deleted 

     Resolution of Figure 7 should be improved ...

Reviewer 2 Report

In this review, the author highlighting the various approaches that have the potential to contribute to ards sustainable sludge management through the enhancement of energy and resource recovery from sewage sludge. On the whole, the review is of considerable interest and well done. I recommend it to be published after a minor revision.

1. The Authors should also proofread their manuscript (some spelling and grammar errors).

2. The conclusion is too long and also not targeted to the important aspects described in the manuscript; please rephrase it.

3. The author should better improve the beauty and quality of the figures and add error bar in Fig. 2.

4. In the introduction part, Some publications are suggested to refer to improve the quality of the manuscript, such as: https://doi.org/10.1016/j.surfin.2022.102006, https://doi.org/10.1016/j.colsurfa.2021.127753.

Reviewer 3 Report

The manuscript contains a broad summary of the techniques for sewage sludge treatment. It is very complete and update. The following comments address some aspects to be considered by the authors in the revised manuscript.

  1. Title: From my point of view “sustainable techniques is very general”. There is not information to validate that no-commercial techniques are environmental and economical sustainable.
  2. Abstract/Manuscript: The authors should indicate the difference between sewage and sewage sludge. Are the authors focused only on sewage sludge from an urban wastewater treatment plant? Can the authors considerer a homogeneity of the sludges depending on their origin?
  3. Abstract/Manuscript: If the authors refer to cost treatment some numbers have to be included (e.g. line 48)
  4. I recommend substituting “cremation” by “incineration”.
  5. A characterization of sewage sludge after CAMBI treatment is required. In the case of Bio Thelys technology, the improved associated with this technology should quantified.
  6. A summary table with a description of the technologies and main variables is requested.
  7. The dimension of digesters (scales) has to be included.
  8. Can the authors indicate the minimum lipids concentration in the sewage sludge to obtain a biodiesel by a sustainable process?
  9. Section 3.1.1 appears as an enumeration of case studies. From my point of view, the quality of the paper would improve if the authors could write a discussion. Again, a summary table is required.
  10. Has the authors reviewed some papers about the biodiesel production from hydrothermal liquefaction?
  11. Table 1 could be completed with characterization of biodiesel obtained by technologies previously mentioned.
  12. Table 4 should include the heavy metal content.
  13. Problems associated with the digestate management are not quantified.
  14. Finally, the authors should include a general overview giving their technical opinion on the best approach to sewage sludge valorization: energy or resource recovery? When one is better than other? Why?

Typo errors:

-          Revise superindex (m3)

-          No plural in units (bars)

Reviewer 4 Report

              This review article adequately presents recent advancements in commercial and other sustainable techniques for energy an material recovery from sewage sludge. Emphasis is placed on anaerobic digestion producing biogas and taking advantage of the high lipid content through esterification and direct sludge pyrolysis to produce biodiesel and bio-oils. Heavy metal recovery from sewer sludge is also discussed in addition to other techniques.

Grammar /Format Mistakes:

Line 43, m3 should be m3

Line 257, m3 should be m3

Line 258, m3 should be m3

Line 448, still has a Track Changes suggestion not accepted.

Line 794, ‘to summaries’ should be to summarize.

Line 57 is the word against missing? … protect ‘against’ sludge landfilling.

After the above corrections are addressed, the article is suitable for publication; however, I offer the following suggestions to the authors to incorporate to strengthen the manuscript if they so desire.

(1) In lines 42-45 the stats are given for how many million tons of sewage sludge is produced each year from the EU, US and China. Given the populations of these, and the numbers provided in these lines of text, it suggests China is doing something incredibly correct. It would be advantageous to comment on the numbers and state how much sewer sludge is actually treated in each of these and the percentages of each that have access to sanitation.

(2) For such a paper, I think it is okay to highlight the challenges with biodiesel use, in an effort for the scientific community to remind them to address these challenges in their research; some of these challenges include simply basic long-term storage (to hot it spoils, to cold it changes its viscosity).

(3) Likewise, for a such a paper, as the discussion of biofuels starts on line 294, are there any studies done to include biodiesel longterm studies where engine damage or lack there of are carried out when using the highest grade biodiesels currently available? There is some reference to carbon depositions in line 354, but can a better case be made for long-term studies with details provided in the review to show more the potential of biodiesels?

(4) Current biodiesels do not meet the 14 parameters in the ASTM D-6751 specifications nor the 16 specifications in EN-14214. This is a significant number of parameters to meet. Could references be included that demonstrate newly designed engines (with or without modification) that demonstrate the potential of biodiesel in its current state (with long terms studies on those engines if possible).

(5) line 609, the reference is provided but can the authors provide more details in the text about the main sources of heavy metals enter into the wastewater treatment processes.

Reviewer 5 Report

General remarks

In the paper many information appear many times, often not directly related to the topic, but there is no technical information related to the processes that are discussed and one can get the impression that they are mixed.

The structure of the article is unclear, sutitles are inconsistent, e.g. in chapter 2, information not related to this subtitle is presented first, then three technologies are presented, then subtitle Currect status of biogas, then again information about devices, e.g. micro scale digesters etc. The idea of such information presentation is incomprehensible to the reader. Same for other chapters.

On the basis of the presented information and conclusions, the reader may get the impression that the main idea of the article is to present and advertise three technologies, there is no presentation of advantages and disadvantages and no comparison with other technologies.

Datailes information

1.     Based on which legislation authors state that incineration of sewage sludge is banned?

2.     line 43 – tons of dry or wet mass?

3.     Line 58-67 information not connected with topic.

4.     In lines 107 -108 authors claim that agricultural application of sewage sludge is friendly for environment but in line 52-53 is presented the opposite idea.

5.     Line 125-134 it is repetition of previously presented information from 136 to 160  the same situation.

6.     Why is the reason of presenting energy needs of Ukraine and Israel?

7.     No reference in the text to Figure 3

8.     What kind of novelty is presented on this Figure 5?

9.     What is the reason of presenting Table 1 and Table 2. There is not any compering with biodiesel obtained from sewage sludge?

10.  Have authors permission of using graphics, ie. Fig. 7 done by other authors?

11.  Title of section 3.3. Process economy.  which process biodiesel or bioil?

12.  Conclusions are so obvious.

Round 2

Reviewer 1 Report

All the comments are well addressed, it can be accepted for publication now

Reviewer 3 Report

After reviewing, the quality of the manuscript has increased. However, some relevant questions have not been taking into account for the authors:

 2. Abstract/Manuscript: The authors should indicate the difference between sewage and sewage sludge. Are the authors focused only on sewage sludge from an urban wastewater treatment plant? Can the authors considerer a homogeneity of the sludges depending on their origin?

The authors describe sewage sludge but they do not clarify the sense of “sewage”.

 6. A summary table with a description of the technologies and main variables is requested.

From my point of view, to include the information in a Table is more attractive for the readers.

12. Table 4 should include the heavy metal content.

The heavy metal content is required to determine if a material could be used as fertilizer (see the law).

Author Response

Please se the attachment
